# The His-Gly motif of acid-sensing ion channels resides in a reentrant 'loop' implicated in gating and ion selectivity

**Nate Yoder[1†], Eric Gouaux[1,2]\***

[1]Vollum Institute, Oregon Health & Science University, Portland, United States; [2]Howard Hughes Medical Institute, Oregon Health & Science University, Portland, United States

**Abstract** Acid-sensing ion channels (ASICs) are proton-gated members of the epithelial sodium channel/degenerin (ENaC/DEG) superfamily of ion channels and are expressed throughout the central and peripheral nervous systems. The homotrimeric splice variant ASIC1a has been implicated in nociception, fear memory, mood disorders and ischemia. Here, we extract full-length chicken ASIC1 (cASIC1) from cell membranes using styrene maleic acid (SMA) copolymer, elucidating structures of ASIC1 channels in both high pH resting and low pH desensitized conformations by single-particle cryo-electron microscopy (cryo-EM). The structures of resting and desensitized channels reveal a reentrant loop at the amino terminus of ASIC1 that includes the highly conserved 'His-Gly' (HG) motif. The reentrant loop lines the lower ion permeation pathway and buttresses the 'Gly-Ala-Ser' (GAS) constriction, thus providing a structural explanation for the role of the His-Gly dipeptide in the structure and function of ASICs.

**\*For correspondence:** gouauxe@ohsu.edu

**Present address:** [†]Department of Physiology, University of California, San Francisco, San Francisco, United States

**Competing interests:** The authors declare that no competing interests exist.

## Introduction

In mammals, four ASIC genes, in concert with splice variants, encode for at least six distinct subunits that assemble as proton-gated, voltage-insensitive heteromeric or homomeric channels (*Deval et al., 2010*). The homotrimeric splice variant, ASIC1a, is found on the dendrites, post-synaptic spines, and cell bodies of central neurons (*Waldmann et al., 1997*; *Zha et al., 2006*) and is enriched in the amygdala (*Wemmie et al., 2003*). ASIC1a channels participate in multiple central nervous system (CNS) processes including fear conditioning (*Chiang et al., 2015*; *Coryell et al., 2008*; *Wemmie et al., 2004*), nociception (*Bohlen et al., 2011*; *Krishtal and Pidoplichko, 1981*) and synaptic plasticity (*Wemmie et al., 2002*; *Du et al., 2014*; *Liu et al., 2016*; *Diochot et al., 2012*). ASICs are also therapeutic targets (*Hu et al., 2011*; *Yin et al., 2013*; *Xiong et al., 2006*), with localization patterns, $Ca^{2+}$ permeability and proton-dependent activation implicating these channels in acidosis-induced neuronal injury (*Wang et al., 2015*; *Duan et al., 2011*; *Xiong et al., 2004*; *Pignataro et al., 2007*; *Wang et al., 2020*) and mood disorders (*Coryell et al., 2009*; *Pidoplichko et al., 2014*). Upon activation by rapid exposure to low pH, homotrimeric ASIC1a channels open, exhibiting modest $Na^+$ selectivity with $P_{Na}/P_K \sim 7.8$ and $P_{Na}/P_{Ca} \sim 18.5$ (*Bässler et al., 2001*; *Yermolaieva et al., 2004*) and subsequently entering a long-lived desensitized state in hundreds of milliseconds (*Hesselager et al., 2004*). A simple gating mechanism consistent with the observed kinetic measurements is comprised of high pH resting, low pH open and low pH desensitized states (*Zhang et al., 2006*; *Gründer and Pusch, 2015*).

ASICs, and by extension, members of the ENaC/DEG superfamily of ion channels, are trimers, with each subunit composed of large extracellular domains, two transmembrane (TM) helices, and intracellular amino and carboxy termini (*Jasti et al., 2007*; *Noreng et al., 2018*). Chicken ASIC1, an ortholog of mammalian ASIC1a, is, from a structural standpoint, the most well-characterized

member of the ENaC/DEG family of ion channels, with X-ray and single-particle cryo-EM structures of the detergent-solubilized channel determined in the resting (*Yoder et al., 2018a*), open (*Baconguis et al., 2014*) and desensitized (*Jasti et al., 2007*; *Gonzales et al., 2009*) states. These structures, defined by conducting and non-conducting pore profiles, 'expanded' and 'contracted' conformations of the thumb domain, and distinct conformations of critical β-strand linkers (*Jasti et al., 2007*; *Yoder et al., 2018a*; *Baconguis et al., 2014*; *Gonzales et al., 2009*; *Baconguis and Gouaux, 2012*; *Dawson et al., 2012*), have provided the foundation for structure-based mechanisms for proton-dependent gating. However, in all these structures, the intracellular amino terminal residues Met 1 through Arg 39, including the highly conserved HG motif, are disordered and not visible in the X-ray diffraction or cryo-EM density maps, leaving a gap in our understanding of how these regions contribute to channel structure and function.

Numerous lines of evidence indicate that the amino terminus of ENaC/DEG channels, which includes the conserved HG motif, contributes to gating and ion conduction properties. Indeed, mutation of the conserved glycine in the β subunit of ENaC channels reduces channel open probability (*Grunder et al., 1997*; *Gründer et al., 1999*) and underlies one form of pseudohypoaldosteronism type 1 (PHA-1) (*Chang et al., 1996*). In ASICs, pre-TM1 residues participate in ion selectivity (*Coscoy et al., 1999*) and proton-dependent gating (*Pfister et al., 2006*) as well as $Ca^{2+}$-permeability (*Bässler et al., 2001*), further demonstrating that the amino terminus of ASIC/ENaC/DEG channels plays an important role in ion channel function and may comprise portions of the ion pore. Despite the wealth of structural information surrounding ASICs and, more recently, the ENaC structure (*Noreng et al., 2018*), the architecture of the cytoplasmic terminal domains as well as the molecular mechanisms by which the HG motif and pre-TM1 residues contribute to channel structure and function have remained elusive.

Here, we present structures of cASIC1 solubilized without the use of detergent, using SMA copolymers (*Knowles et al., 2009*; *Dörr et al., 2014*), in distinct conformational states at low and high pH. Our results reveal that amino terminal pre-TM1 residues form a reentrant loop and that the HG motif is situated 'below' the GAS belt, TM2a/b domain swap, at a subunit interface and along the lower ion permeation pathway. Furthermore, we show that the lower half of the ion permeation pathway in resting and desensitized states is comprised entirely of pre-TM1 reentrant loop residues, informing mechanisms for the contribution of the amino terminus to ion permeation properties. Finally, we observe lipid-like density features surrounding the transmembrane domain (TMD) that suggest preservation of protein-lipid interactions by the SMA-mediated detergent-free isolation methods.

## Results

### Isolation and structure determination of cASIC1 in SMA copolymer

To elucidate structures of cASIC1 bound with endogenous lipids, we extracted and purified recombinant channels in the presence of SMA copolymer. After a two-step chromatographic purification procedure, cASIC1-SMA protein was ~95% pure as judged by SDS-PAGE and was monodisperse as measured by fluorescence-detection size exclusion chromatography (FSEC) (*Kawate and Gouaux, 2006*; *Figure 1—figure supplement 1A–B*). Negative stain transmission electron microscopy also demonstrated good particle distribution and limited aggregation (*Figure 1—figure supplement 1C*).

We next pursued single-particle cryo-EM of cASIC1-SMA, obtaining reconstructions of ASIC1 channels in low pH desensitized (pH 7.0) and high pH resting (pH 8.0) conformations at estimated resolutions of ~2.8 and 3.7 Å, respectively, as estimated by gold-standard FSC (*Rosenthal and Henderson, 2003*; *Figure 1A–B*, *Supplementary File 1*, *Figure 1—figure supplements 2–5*). While $pH_{50}$ for cASIC1 is ~6.7 (29, 31), extensive 3D classification of the pH 7.0 dataset did not indicate the presence of either open or resting channels. This observation is consistent with previous electrophysiological analysis of steady-state desensitization curves for cASIC1, which demonstrated very little proton-evoked current after conditioning with ~pH 7.0 solution (*Yoder et al., 2018a*). Accordingly, we speculate that our cryo-EM reconstruction is representative of the long-lived desensitized state occupied by cASIC1 channels following extended exposure to low, but sub-threshold, pH conditions (*Babini et al., 2002*).

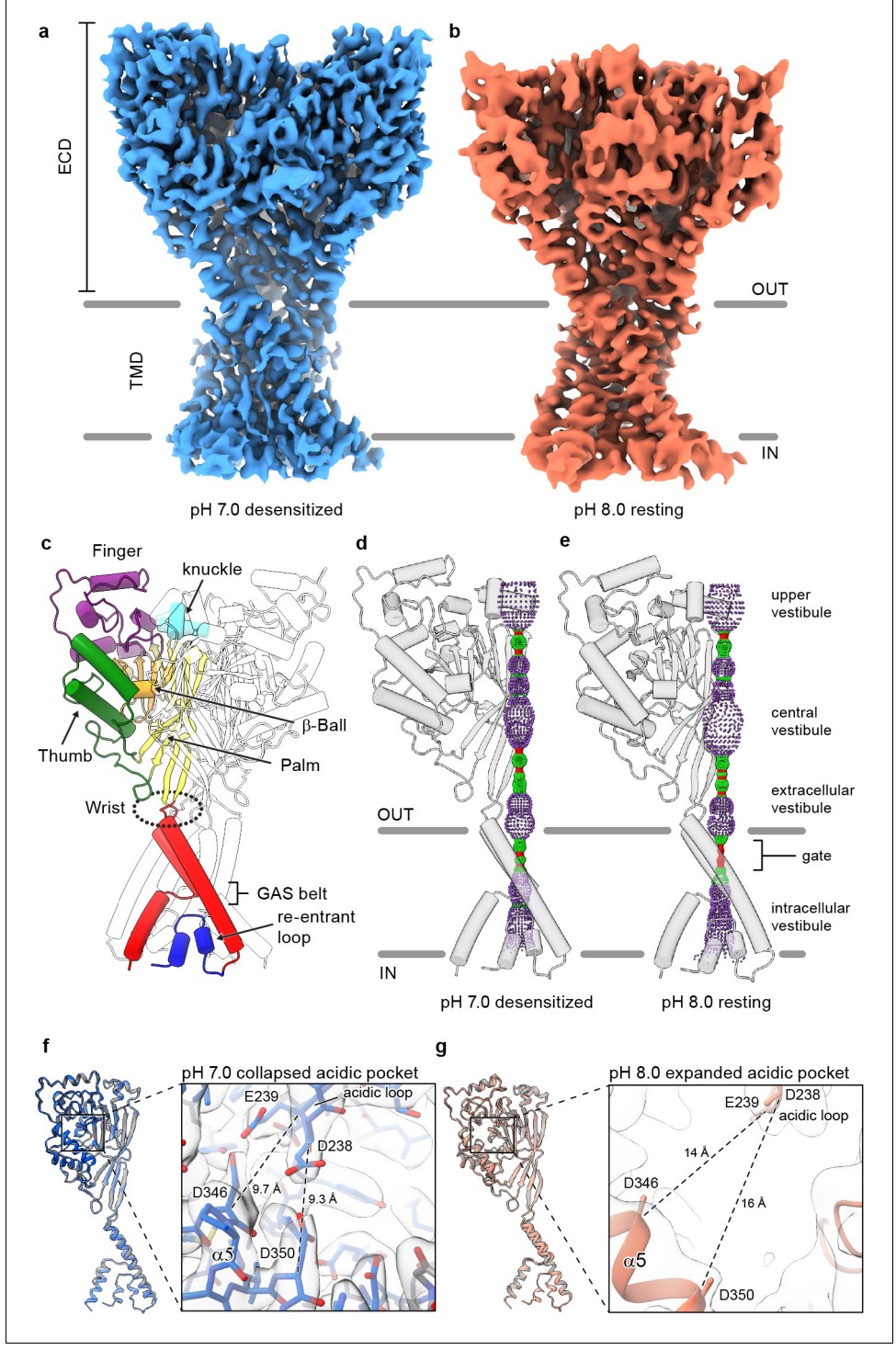

**Figure 1.** Structures of cASIC1-SMA. (**a-b**) Cryo-EM maps of cASIC1-SMA at pH 7.0 (**a**) and pH 8.0 (**b**). (**c**) Cartoon diagram of cASIC1 with single subunit shown colored by domain. (**d-e**) Pore profiles for cASIC1-SMA in a desensitized (**d**) and resting (**e**) state calculated with HOLE software (pore radius: red <1.15 Å<green < 2.3 Å<purple). (**g**) Single subunit superposition of cASIC1-SMA in a desensitized state (blue) and the desensitized state X-ray structure (*Baconguis et al., 2014*; *Gonzales et al., 2009*) (PDB 4NYK, grey). Detailed view of the collapsed acidic pocket is shown in the inset. (**h**) Single subunit superposition of cASIC1-SMA in a resting state (salmon) and the resting state X-ray structure (*Yoder et al., 2018a*) (PDB 5WKU, grey). Detailed view of the expanded acidic pocket is shown in the inset.

*Figure 1 continued on next page*

*Figure 1 continued*

The online version of this article includes the following figure supplement(s) for figure 1:

**Figure supplement 1.** Purification of cASIC1-SMA.
**Figure supplement 2.** Cryo-EM of cASIC1-SMA at pH 7.0.
**Figure supplement 3.** Cryo-EM data processing for cASIC1-SMA at pH 7.0.
**Figure supplement 4.** Cryo-EM of cASIC1-SMA at pH 8.0.
**Figure supplement 5.** Cryo-EM data processing for cASIC1-SMA at pH 8.0.

In accord with previously solved structures, the homotrimeric ASIC1 channel resembles a clenched fist (*Jasti et al., 2007*), harboring domain-swapped TM2 helices (*Baconguis et al., 2014*; *Figure 1C*). Both the desensitized and resting channels adopt closed ion channel gates, as evidenced by a constriction between residues 433–436 within the upper third of the TMD (*Figure 1D–E*). At pH 7.0, cASIC1-SMA particles prepared at pH 7.0 populate a desensitized state that mirrors the overall architecture of the existing X-ray structure (*Gonzales et al., 2009*), including the presence of a proton-bound 'collapsed' acidic pocket (*Figure 1F*). In contrast, cASIC1-SMA particles maintained at pH 8.0 occupy a high pH resting conformation, characterized by an expanded acidic pocket (*Figure 1G*) that resembles the high pH, resting state structures solved by X-ray crystallography and single particle cryo-EM (*Yoder et al., 2018a*). We propose that the limited resolution of the resting channel structure, while presumably impacted by cryo-EM grid conditions, including thicker ice than for the pH 7.0 grids, may also be due to structural flexibility inherent to the resting channel conformation in the absence of divalent cations, which serve to stabilize an expanded acidic pocket at high pH (*Yoder and Gouaux, 2018b*) but which are incompatible with current SMA-based purification strategies.

## Amino terminal residues form a reentrant loop

Numerous experiments have implicated residues within the pre-TM1 region of ASICs and ENaCs in both gating and selectivity (*Grunder et al., 1997*; *Gründer et al., 1999*; *Coscoy et al., 1999*). Indeed, the highly conserved HG motif is located within the pre-TM1 region of ASICs and ENaCs and its disruption lowers the open probability in ENaCs and underlies PHA type 1 disorder (*Grunder et al., 1997*; *Chang et al., 1996*). In contrast to existing structures of ASICs solubilized in detergent micelles, we observed strong protein density corresponding to amino terminal residues in cryo-EM maps of both desensitized and resting cASIC1-SMA channels maintained in a lipid environment (*Figure 2A–B*).

The quality of the 2.8 Å density map of the desensitized channel was sufficient to build amino terminal residues into the density map, beginning with Val 17, in the context of the previously determined desensitized state structure (*Baconguis et al., 2014*; *Gonzales et al., 2009*) (PDB 4NYK) (*Figure 1—figure supplement 2*). The pre-TM1 residues, from Val 17 to Leu 40, form a reentrant loop comprised of two short helical segments (Re-1 and Re-2) separated by a turn, positioned on the cytoplasmic side of the GAS belt (*Figure 2C–E*). Interestingly, the presence of the reentrant loop does not noticeably impact the position of either TM helix from those observed in prior X-ray or cryo-EM structures. Rather, the reentrant loop residues are 'pinned' within the inverted 'v-shaped' cavity formed between the lower TM helices and maintained primarily by virtue of intra-subunit contacts with TM2b and TM1 (*Figure 2—figure supplement 1*).

While the quality of the resting channel density map that includes the pre-TM1 residues was not sufficient for unambiguous model building (*Figure 1—figure supplement 4*), no significant differences in reentrant loop conformation were observed between the desensitized or resting channels at the current resolutions (*Figure 2F*), allowing us to rigid body fit the pre-TM1 structural element derived from the desensitized state structure into the resting state map. In contrast with the discovery of new density for the pre-TM1 region, we neither observed interpretable density associated with the carboxy terminus, nor for amino terminal residues preceding Val 17, in either the desensitized or resting state maps, thus suggesting that even in SMA-solubilized protein, these regions are disordered.

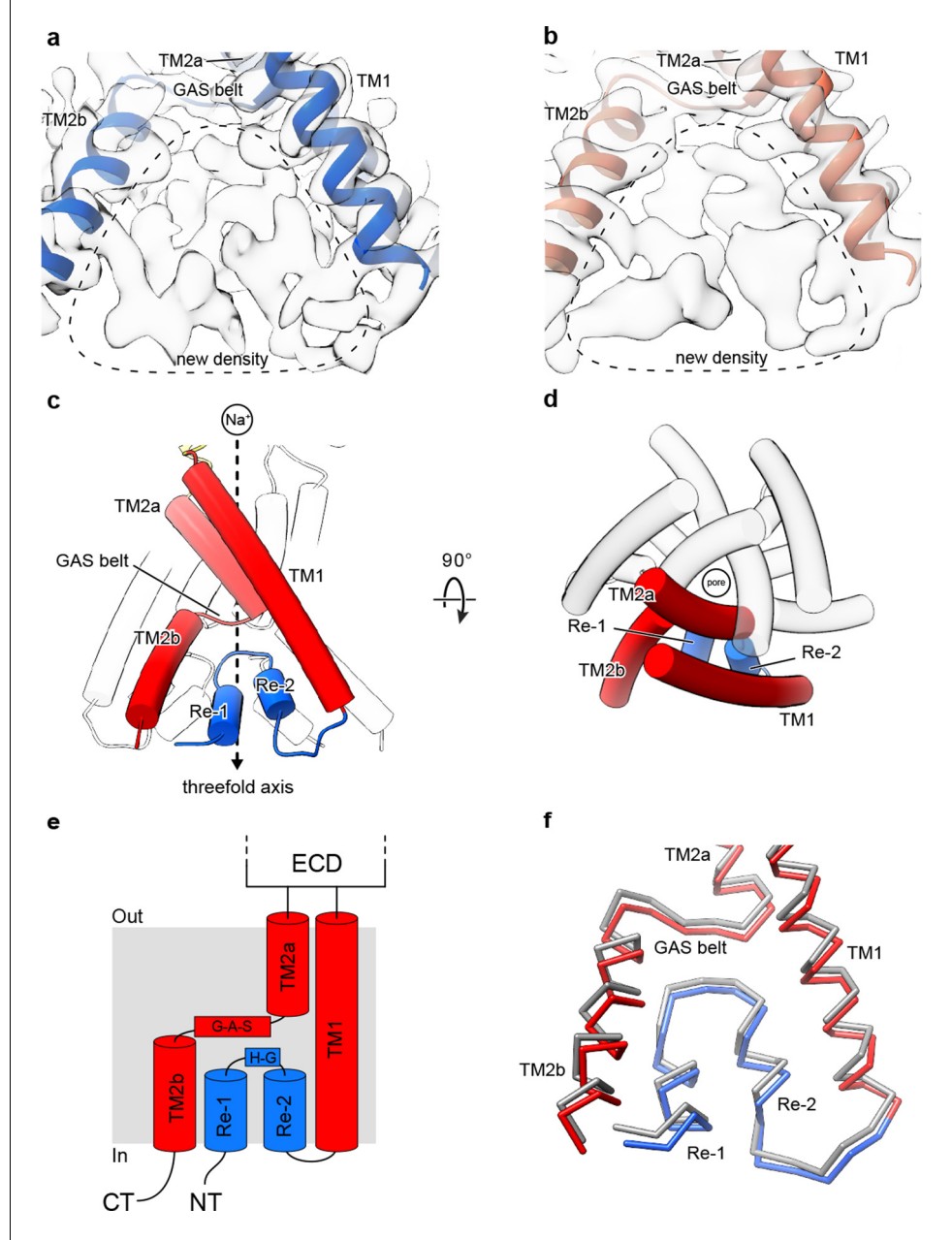

**Figure 2.** Cryo-EM density and structure of the reentrant, pre-TM1 domain. (a-b) Cryo-EM density corresponding to amino terminal residues of cASIC1-SMA at pH 7.0 (a) and pH 8.0 (b). (c–d) Side (c) and top-down (d) views of the TMD from cASIC1-SMA in a desensitized state at low pH. A single subunit is shown, colored by domain. (e) Schematic depicting the TMD topology of cASIC1 channels. (f) Backbone superposition of cASIC1-SMA in the desensitized (colored by domain) and resting (grey) states.

The online version of this article includes the following figure supplement(s) for figure 2:

**Figure supplement 1.** Polar contacts at the reentrant loop.

## The reentrant loop harbors the HG motif

Separated by more than 400 residues in the amino acid sequence, the GAS and HG motifs are highly conserved amongst ENaC/DEG and ASIC channels (*Figure 3A*) and have been implicated in gating (*Gründer et al., 1999*) and ion selectivity (*Kellenberger et al., 1999a*; *Kellenberger et al., 1999b*; *Kellenberger and Schild, 2002*; *Kellenberger et al., 2001*). Interestingly, the HG motif, which

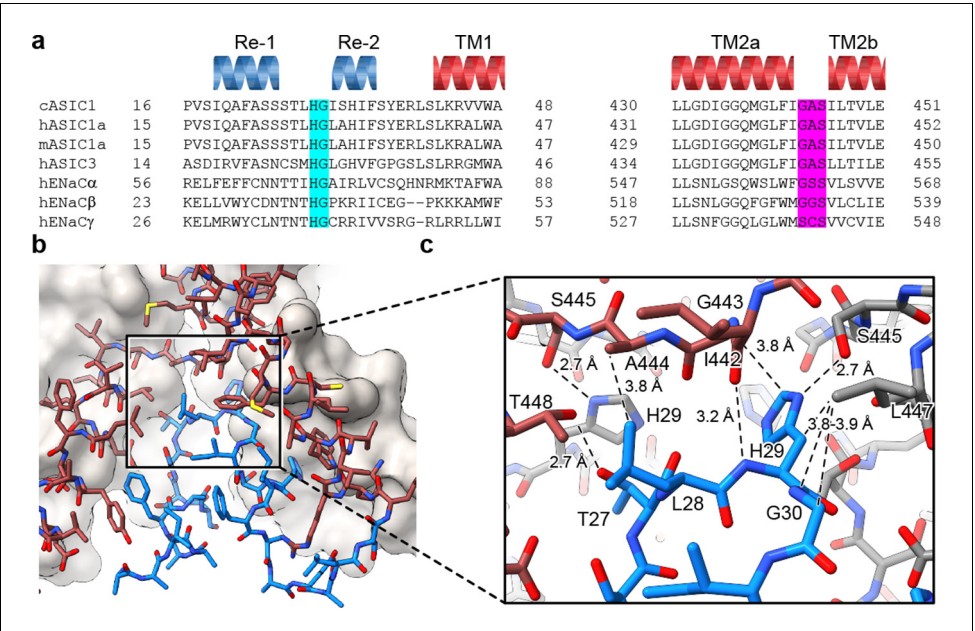

**Figure 3.** The HG motif resides at a subunit interface 'below' the GAS belt. (a) Sequence alignment of selected ASIC1, ASIC3 and ENaC channels covering the pre-TM1 and TM2 domains with GAS domain and HG motif residues highlighted in pink and light blue, respectively, and with secondary structure for cASIC1 shown above the sequences. (b–c) View of the chemical environment around the reentrant loop (b) with a detailed view of the GAS domain and HG motif interface (c).

contains a well-characterized disease mutation in ENaCs at the universally conserved glycine residue (*Grunder et al., 1997*; *Chang et al., 1996*), is situated on the turn between the reentrant helices where it buttresses the TM2a/b domain swap and GAS belt residues from 'below' (*Figure 3B*). Residing along the ion permeation pathway and at a subunit interface, the HG motif is capped by the carboxy terminus of TM2a via an intra-subunit hydrogen bonding interaction with Ile 442 and participates in an inter-subunit hydrogen bonding interaction with a neighboring GAS belt residue via Ser 445 (*Figure 3C*). This intricate network of intra- and inter-subunit interactions, formed between highly conserved motifs via the TM2 domain swap and amino terminal reentrant loop, is consistent with the lower pore architecture playing an important role in ASIC, and by extension, in ENaC function.

## Pre-TM1 residues form the lower ion permeation pathway

In structures of both desensitized and resting cASIC1-SMA channels, the 'upper' ion permeation pathway is comprised of TM2a residues and contains a closed gate between Gly 432 and Gly 436, in agreement with existing X-ray and cryo-EM models (*Figure 4A*). However, where structures of ASIC1 in resting (*Yoder et al., 2018a*), open (*Baconguis et al., 2014*) and desensitized (*Gonzales et al., 2009*) conformations highlight a 'lower' ion permeation pathway comprised entirely of TM2b residues that expands outwards to form a wide intracellular vestibule, pre-TM1 residues of the SMA-isolated cASIC1 channels line a more narrow ion permeation pathway extending below the GAS belt (*Figure 4A–B*).

The 'lower' ion conduction pathway of resting and desensitized cASIC1-SMA channels is formed by reentrant amino terminal residues Ser 24 through His 29 (*Figure 4C*), the latter of which is situated below the GAS belt and is oriented towards the threefold axis where it forms a constriction below the gate in the desensitized channel (*Figure 4D*). Our data demonstrate that pre-TM1 residues line the lower ion conduction pathway in structures of resting and desensitized cASIC1 channels, providing a structural rationale for earlier reports which indicated that pre-TM1 residues may

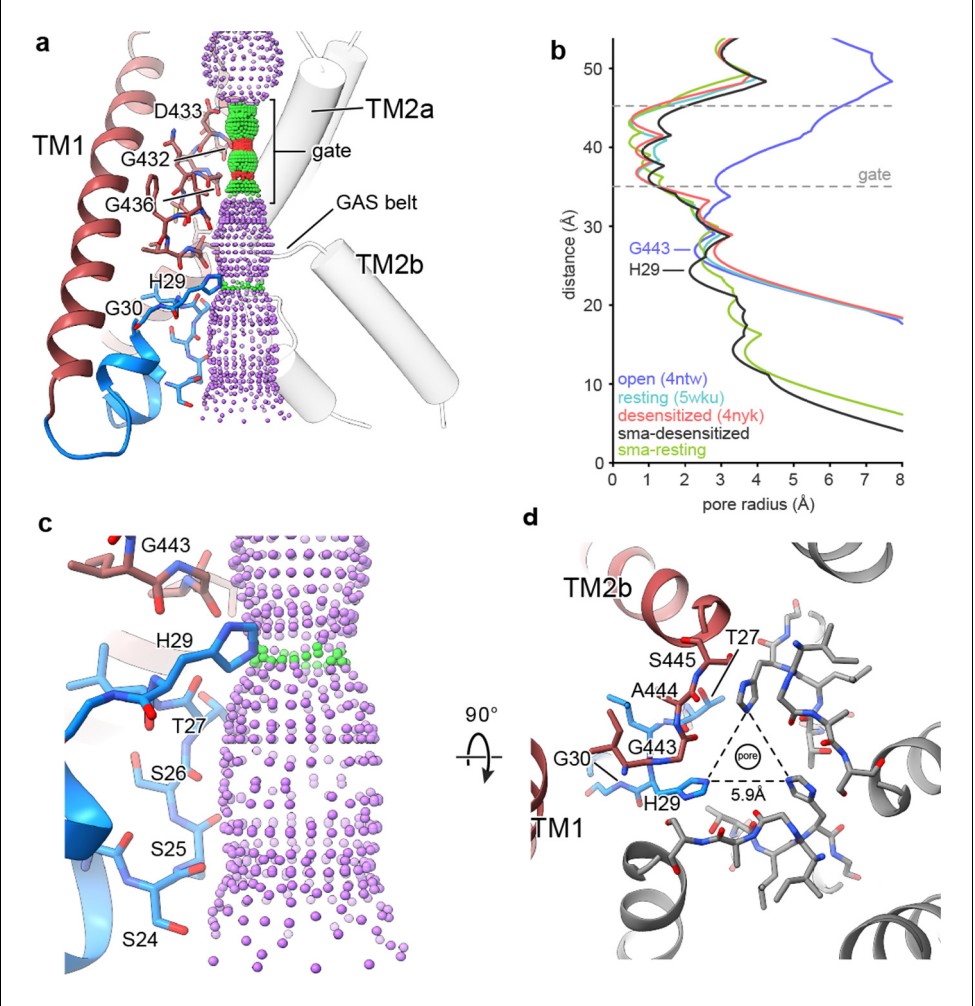

**Figure 4.** The reentrant loop forms the lower ion permeation pathway. (a) Overview of pore-forming residues of desensitized cASIC1-SMA channels beginning at the ion channel gate. Pore profile calculated with HOLE software (pore radius: red <1.15 Å<green < 2.3 Å<purple) is shown. (b) Plot of pore radius corresponding to the view in (a) for resting (*Yoder et al., 2018a*) (PDB 5WKU), open (*Baconguis et al., 2014*) (PDB 4NTW) and desensitized (*Baconguis et al., 2014*; *Gonzales et al., 2009*) (PDB 4NYK) X-ray and cASIC1-SMA cryo-EM structures. (c) Detailed view of the 'lower' ion permeation pathway formed by pre-TM1 residues. (d) Top-down view of the constriction 'below' the ion channel gate formed by His 29, as visualized in the desensitized cASIC1-SMA state. The GAS belt residues are in the foreground.

form part of the pore (*Pfister et al., 2006*) and contribute to ion permeation (*Bässler et al., 2001*) and $Na^+$ selectivity of ASICs (*Coscoy et al., 1999*).

In the X-ray structure of an open channel conformation, hydrated $Na^+$ ions encounter a constriction at the GAS belt TM2 domain swap (*Baconguis et al., 2014*) which has long been thought to underpin ion selectivity in ENaC/DEG channels (*Kellenberger et al., 1999a*; *Kellenberger and Schild, 2002*; *Snyder et al., 1999*). Recently, however, residues along TM2a and TM2b both 'above' and 'below' the GAS belt have been demonstrated to be important determinants of selectivity in ASIC1a (*Lynagh et al., 2017*). Despite the presence of an ordered reentrant loop and a narrower pore, we did not observe a change in the position of either TM2a/b or the GAS belt residues between the resting and desensitized conformations. Additional studies of the open state, perhaps under SMA isolation conditions, will be required to illuminate the structure of the activated, ion conducting state of the channel.

## Density features suggest TMD-lipid interactions

The reconstitution of membrane proteins into lipid nanodiscs is a well-established technique in structural biochemistry that permits the study of sensitive membrane proteins embedded in phospholipid bilayers (*McLean et al., 2018*; *Bayburt et al., 1998*). While a reconstitution approach provides for a controlled and defined lipid environment, the necessity of an initial detergent-based extraction step may disrupt protein-lipid interactions integral to the structural integrity of TM segments. In contrast with nanodisc reconstitution, SMA copolymers extract membrane proteins directly from the lipid bilayer, permitting the study of membrane proteins in the presence of endogenous lipids (*Esmaili and Overduin, 2018*; *Sun et al., 2018*) and, in principle, maintaining the native protein-lipid interactions (*Teo et al., 2019*).

In our 2.8 Å reconstruction of a desensitized ASIC1, we observed multiple ordered elongated densities situated in hydrophobic channels along the TMD (*Figure 5A*) that we suggest may correspond to bound lipids. Separated into spatially distinct clusters (*Figure 5B*), putative lipid densities

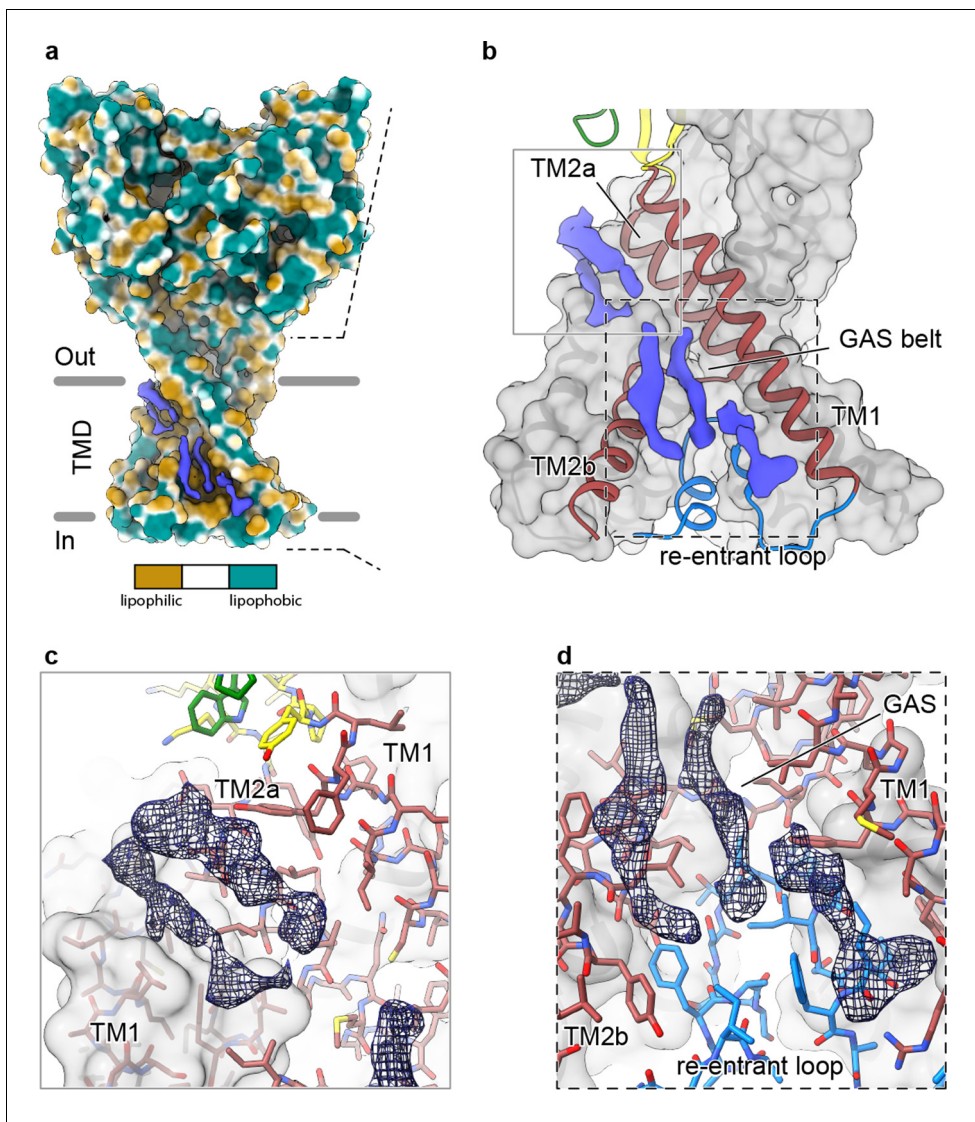

**Figure 5.** Elongated density within lipophilic channels at the TMD of cASIC1-SMA channels. (a) Surface representation of cASIC1-SMA in the desensitized state colored by lipophilicity potential calculated with pyMLP (*Laguerre et al., 1997*) in ChimeraX (*Goddard et al., 2018*). (b) Hybrid cartoon and surface representation of putative lipid sites at the TMD. (c–d) Putative lipid densities between TM1 and TM2a (c) and between TM1 and TM2b adjacent to the reentrant loop (d) of cASIC1-SMA.

reside near the top of the membrane sandwiched between TM2a and TM1 helices (*Figure 5C*) and near the cytoplasmic side of the membrane, between TM1 and TM2b (*Figure 5D*).

Our results suggest that the local lipid environment is important for maintaining the architecture of the reentrant pre-TM1 residues in ASIC1a and thus the integrity of the lower pore pathway. Therefore, the location of at least one cluster of putative lipid densities within a lipophilic cleft adjacent to reentrant loop residues and the GAS belt (*Figure 5D*) is intriguing, especially given that the cryo-EM structure of a full-length cASIC1 channel in *n*-dodecyl-β-D-maltoside (DDM) lacked ordered amino terminal residues (*Yoder et al., 2018a*). However, given the resolution of our cASIC1-SMA reconstructions, we are unable to assign this density to any specific lipid. Future experiments are needed to determine the molecular composition of cASIC1-SMA particles and to explore relevant interactions between ASICs and the plasma membrane.

## Discussion

Here, we present structures of chicken ASIC1 solubilized by SMA in high pH resting and low pH desensitized conformations. While the conformation of both resting and desensitized channels throughout the ECD faithfully mirrors those solved previously via detergent-based methods (*Yoder et al., 2018a*; *Gonzales et al., 2009*), our structures demonstrate that amino terminal residues prior to TM1 form a reentrant loop that comprises the lower portion of the ion permeation pathway. In both resting and desensitized structures, the conserved HG motif is situated within the reentrant loop, immediately below the GAS belt, TM2 domain swap, where the imidazole group of the His forms a constriction along the ion permeation pathway and is stabilized by a complex network of inter- and intra-subunit interactions. Finally, we detected elongated ordered densities within lipophilic channels of the TMD, some of which are adjacent to the reentrant amino terminus, that may correspond to bound lipids.

Our cryo-EM reconstructions of full-length cASIC1 demonstrate that SMA-solubilized channels occupy a resting state at high pH and enter a desensitized state following exposure to pH 7.0. These results are in good agreement with both prior structural studies of chicken ASIC1 (*Yoder and Gouaux, 2018b*; *Gonzales et al., 2009*) and a simple three state model for proton-dependent gating (*Zhang et al., 2006*; *Gründer and Pusch, 2015*), indicating that isolated cASIC1 channels solubilized by SMA retain proton-dependent gating properties. Accordingly, we propose that these structures are representative of proton-gated channels in physiologically meaningful conformations. Nevertheless, we acknowledge that these studies do not directly address the function of cASIC1-SMA channels in isolation, nor do they define the contribution of individual residues of the pre-TM1 reentrant loop to gating and selectivity, and thus additional experiments are required to fully elucidate the mechanistic and functional implications of our structural observations.

X-ray structures of chicken ASIC1 deletion mutants have highlighted significant conformational heterogeneity at the TMD, complicating the interpretation of how TMD architecture relates to gating and ion selectivity in ASICs (*Jasti et al., 2007*; *Yoder et al., 2018a*; *Baconguis et al., 2014*; *Gonzales et al., 2009*; *Baconguis and Gouaux, 2012*; *Dawson et al., 2012*). While domain-swapped TM2 helices and GAS belt constrictions are now established structural characteristics of functional channels in desensitized (*Baconguis et al., 2014*; *Gonzales et al., 2009*), toxin-stabilized open (*Baconguis et al., 2014*), and resting (*Yoder et al., 2018a*) conformations, X-ray structures of ASIC1 channels bound to the gating modifier toxin, PcTx1 (*Baconguis and Gouaux, 2012*), harbor continuous TM helices, indicating that a subset of conformational states outside of the canonical proton-dependent gating pathway may deviate from the domain-swapped architecture. While the capacity of ASICs to adopt both domain-swapped and continuous TM2 helices is supported by both X-ray structures and evolutionary analysis (*Kasimova et al., 2020*), the functional relevance of a 'dual-mode' lower pore architecture remains enigmatic.

In light of the structural heterogeneity at the lower pore of ASICs, the presence of lipid-like densities along the reentrant loop of cASIC1-SMA channels is particularly notable. Indeed, these studies represent the first structures of an ASIC not solubilized by detergent. We speculate that a lipid-like environment, maintained by virtue of detergent-free, SMA-mediated protein extraction and purification methods, may be important for maintaining the integrity of the lower pore structure of ASIC1 channels. Accordingly, structures of the cASIC1-SMA complex provide evidence of an ordered pre-TM1 reentrant loop in the high pH resting and low pH desensitized conformations. Though

speculative, a requirement for lipid cofactors in protecting the pre-TM1 reentrant loop would explain why this unique domain architecture was not observed in either the cryo-EM structure of full-length channel nor in any of the prior X-ray structures, all of which represent channels solubilized in DDM.

The mechanism of $Na^+$ selectivity in ENaC/DEG channels has been a topic of intense scrutiny. Early studies demonstrated that the profound $Na^+$-selectivity of ENaCs arises from residues within the highly conserved G/S-X-S motif of TM2 (*Kellenberger et al., 1999a*; *Kellenberger et al., 2001*; *Snyder et al., 1999*; *Sheng et al., 2000*; *Sheng et al., 2001*), a hypothesis further supported by functional studies of ASIC1a channels (*Carattino and Della Vecchia, 2012*; *Li et al., 2011*) as well as the observation that, in ASICs, these residues form a narrow constriction within the pore of the open channel (*Baconguis et al., 2014*). In contrast, recent studies of ASIC1a homomers (*Lynagh et al., 2017*; *Lynagh et al., 2020*) and ASIC1a/2a (*Lynagh et al., 2020*) heteromers have shown that negatively charged residues on TM2b may contribute more to ion selectivity than GAS belt residues. Our structures of SMA-solubilized cASIC1 in resting and desensitized states indicate that TM2b residues do not make direct contributions to the ion permeation pathway and, as such, these observed impacts on selectivity may instead be due to an indirect effect such as a destabilization of the 'lower' pore. Nevertheless, we acknowledge the possibility that the structure of a full-length ASIC1 channel in a lipid-like environment and in a proton-activated open state may deviate from the existing X-ray structure of a toxin-stabilized open channel such that TM2b residues may indeed be in a position to directly influence selectivity.

Prior functional experiments have demonstrated roles for pre-TM1 residues in gating (*Pfister et al., 2006*), selectivity (*Coscoy et al., 1999*), and $Ca^{2+}$-permeability (*Bässler et al., 2001*) of ASICs, suggesting that these residues comprise or, at the very least, influence the open ion channel pore. Lacking the structure of an ASIC1 channel in a proton-activated state, we can only speculate on how pre-TM1 reentrant loop residues might influence ion selectivity and proton-dependent gating. Nevertheless, our observation of a second constriction below the GAS belt formed by a pore-facing His residue (His 29), a residue conserved across members of the ENaC/DEG superfamily, is particularly intriguing. While little is known about this His residue in ASICs, mutation of the corresponding residue in ENaCα to Ala, Asn or Cys results in largely diminished currents and substitution to Arg abolishes $Na^+$ currents entirely (*Gründer et al., 1999*; *Kucher et al., 2011*).

Our data suggest a role for His 29 in stabilizing the pre-TM1 reentrant loop and lower pore conformation via hydrogen bonding interactions with GAS belt residues on neighboring subunits. Furthermore, the size of the lower constriction formed by His 29 (radius ~2.1–2.6 Å, *Figure 4B*) is unlikely to accommodate hydrated $K^+$ ions (*Mähler and Persson, 2012*), indicating a potential contribution to ion selectivity by a simple steric mechanism. Despite clear evidence that the conserved His is critical to the function of ENaC/DEG channels (*Gründer et al., 1999*; *Kucher et al., 2011*), however, there is currently no data to support a role in ion selectivity. Therefore, while our results indicate that the conserved His may play a previously unanticipated role in selectivity, we are unable to extend our analysis beyond these general suppositions. In light of these new structural insights as well as the disparity between recent structural and functional results, additional experiments are required to inform a comprehensive understanding of $Na^+$ selectivity in ASICs.

Notably, the ~60 residue carboxy terminus of ASICs has been consistently disordered in cryo-EM structures of both detergent-solubilized and now SMA-solubilized full-length cASIC1 channels in either resting, or resting and desensitized states, respectively. These results stand in contrast to recent structures of another homotrimeric cation channel, the detergent-solubilized full-length P2X₇ receptor, which exposed large, structured intracellular terminal domains that together form a 'cytoplasmic ballast' and prevent desensitization via palmitoylation-dependent interactions with the membrane bilayer (*McCarthy et al., 2019*). Contrary to P2X receptors, the intracellular carboxy terminus of ASIC1a is predicted to have little, if any, secondary structure, does not appear to contribute to channel gating and is instead likely to serve as a nexus for interactions with cytoplasmic proteins (*Zeng et al., 2014*; *Zha, 2013*), thus highlighting the need for future studies to pursue the structure of ASICs in complex with cellular binding partners in an effort to better illuminate the contribution of these residues to ASIC biology.

Our studies of the cASIC1-SMA complex provide a structural basis for contributions of the amino terminus to ion permeation and proton-dependent gating of ASICs, reveal the location of the conserved HG motif along the ion conduction pathway and expose a role for the plasma membrane in maintaining the TMD architecture of ASICs. Given the structural similarities between ENaCs and

ASICs, as well as the highly conserved and functionally-important nature of the HG and GAS belt residues, these results provide detailed structural information pertaining to a pair of motifs central to gating and ion permeation and of possible therapeutic relevance to the superfamily of ENaC/DEG ion channels.

# Materials and methods

**Key resources table**

| Reagent type (species) or resource | Designation | Source or reference | Identifiers | Additional information |
|---|---|---|---|---|
| Gene (*Gallus gallus*) | Acid-sensing ion channel isoform 1 | Synthetic | NCBI Reference Sequence: NP_001035557.1 | |
| Cell line (*Homo sapiens*) | HEK293S GnTI⁻ | DOI: 10.1073/pnas.212519299 | RRID:CVCL_A785 | |
| Recombinant DNA reagent | pEG BacMam | DOI: 10.1038/nprot.2014.173 | | |
| Chemical compound, drug | SL30010 (SMALP 30010 P) | Polyscope | | http://polyscope.eu/markets/polyscience/ |
| Software, algorithm | MotionCor2 | DOI: 10.1038/nmeth.4193 | RRID:SCR_016499 | http://msg.ucsf.edu/em/software/motioncor2.html |
| Software, algorithm | Gctf | DOI: 10.1016/j.jsb.2015.11.003 | RRID:SCR_016500 | https://www.mrc-lmb.cam.ac.uk/kzhang/Gctf/ |
| Software, algorithm | DoG Picker | DOI: 10.1016/j.jsb.2009.01.004 | RRID:SCR_016655 | https://omictools.com/dog-picker-tool |
| Software, algorithm | cryoSparc V2 | DOI: 10.1038/nmeth.4169 | RRID:SCR_016501 | https://cryosparc.com/ |
| Software, algorithm | Bsoft | DOI: 10.1006/jsbi.2001.4339 | RRID:SCR_016503 | https://lsbr.niams.nih.gov/bsoft/ |
| Software, algorithm | Coot | DOI: 10.1107/S0907444910007493 | RRID:SCR_014222 | https://www2.mrc-lmb.cam.ac.uk/personal/pemsley/coot/ |
| Software, algorithm | Phenix | DOI: 10.1107/S2059798318006551 | RRID:SCR_014224 | https://www.phenix-online.org/ |
| Software, algorithm | MolProbity | DOI: 10.1107/S0907444909042073 | RRID:SCR_014226 | http://molprobity.biochem.duke.edu |
| Software, algorithm | Pymol | PyMOL Molecular Graphics System, Schrodinger, LLC | RRID:SCR_000305 | http://www.pymol.org/ |
| Software, algorithm | UCSF Chimera | DOI: 10.1002/jcc.20084 | RRID:SCR_004097 | http://plato.cgl.ucsf.edu/chimera/ |
| Software, algorithm | UCSF ChimeraX | DOI: 10.1002/pro.3235 | RRID:SCR_015872 | http://cgl.ucsf.edu/chimerax/ |

## Expression and purification of cASIC1 channels

Recombinant full-length acid-sensing ion channels (*Gallus gallus*) were expressed in HEK293S GnTI⁻ cells and membrane fractions were isolated as previously described (*Yoder et al., 2018a*). Membrane pellets were resuspended in ice cold Tris-buffered saline (TBS, 20 mM Tris pH 8.0 and 150 mM NaCl) containing protease inhibitors, dispersed using a dounce homogenizer, and solubilized for 1 hr at 4°C by addition of SL 30010 (Polyscope) SMA copolymers to 2% (w/v) final concentration.

Membrane debris was removed via centrifugation at 125,171 rcf for 30 min at 4°C and the supernatant was incubated with Ni-NTA beads overnight at 4°C in the presence of 10 mM imidazole.

The Ni-NTA bead suspension was then transferred to a XK-16 column and subject to two washes, first with three column volumes of TBS containing 10 mM imidazole and last with three column volumes of TBS containing 30 mM imidazole. The cASIC1-SMA protein was eluted with TBS containing 250 mM imidazole, and peak fractions were pooled and concentrated to ~5 mg/ml. The $His_8$ EGFP tag was removed via overnight thrombin digestion at room temperature (RT) using a ratio of cASIC1 to thrombin of 25:1. The following day, the cASIC1-SMA protein was purified via size-exclusion chromatography (Superose 6 10/300) using a mobile buffer composed of TBS supplemented with 1 mM DTT. A single peak fraction was collected and concentrated to ~1 mg/ml for cryo-EM sample preparation.

## Cryo-EM of cASIC1-SMA

Quantifoil holey carbon grids (R1.2/1.3 200 mesh Au) were glow discharged for 1 min at 15 mA, carbon side facing up. For structure determination of cASIC1-SMA particles at high pH, purified protein at ~1 mg/ml was used immediately for grid preparation. For structure determination at low pH, the pH of the sample was adjusted to 7.0 by addition of MES, pH 6.0, following concentration of purified protein to ~1.0 mg/ml. A 4 µl droplet of sample, applied to the carbon side of the grid, was blotted manually with pre-cooled filter paper (Whatman, grade 1) and the grids were vitrified in ethane/propane mix using a custom-built manual-plunge apparatus housed in a 4°C cold room with 60–70% relative humidity.

## Cryo-EM data acquisition for cASIC1-SMA

For the resting channel structure at high pH, data were collected on a Titan Krios cryo-electron microscope (ThermoFisher) operated at 300 keV. Images were recorded on a Gatan K3 camera positioned after an energy filter (20 eV slit width) operating in super-resolution mode with a binned pixel size of 0.648 Å. Data were collected with SerialEM (*Mastronarde, 2003*) and dose-fractionated to 50 frames for a total exposure time of 2–3 s and a total dose of 40–50 e⁻ Å⁻². 

For the desensitized state structure at pH 7.0, data were recorded on a Titan Krios cryo-electron microscope operated at 300 kV and equipped with a spherical aberration corrector. Images were recorded on a Gatan K2 Summit camera in super-resolution mode with a binned pixel size of 1.096 Å. Data were acquired using Leginon (*Suloway et al., 2005*) and dose-fractionated to 48 frames at 0.15 s per frame for a total exposure time of 7.25 s and a total dose of 50 e⁻ Å⁻².

## Cryo-EM data processing for cASIC1-SMA

Images were motion corrected using UCSF MotionCor2 (*Zheng et al., 2017*) and CTF estimation was performed using Gctf (*Zhang, 2016*). Particles picked using DoG Picker (*Voss et al., 2009*) were subjected to reference-free 2D classification in cryoSPARC V2 (*Punjani et al., 2017*). Following initial classification, an *ab-initio* model was generated in cryoSPARC V2 and used for iterative rounds of 3D classification and refinement in cryoSPARC V2. For the pH 7.0 dataset, per-particle CTF estimation was performed using Gctf. Final reconstructions for both datasets were obtained via non-uniform refinement (C3 symmetry) in cryoSPARC V2.

## Model building and refinement for cASIC1-SMA

Using UCSF Chimera (*Goddard et al., 2007*), the X-ray structures for resting (PDB 5WKU) (*Yoder et al., 2018a*) and desensitized (PDB 4NYK) (*Baconguis et al., 2014*; *Gonzales et al., 2009*) channels were docked into cryo-EM density maps corresponding to pH 8.0 and pH 7.0 datasets, respectively. Docked models were used as templates for iterative rounds of manual model building in Coot (*Emsley et al., 2010*) and real-space refinement in Phenix (*Afonine et al., 2018*). The final models contain residues 17–462 of cASIC1 and were validated using MolProbity (*Chen et al., 2010*; *Supplementary file 1*). Secondary structure predictions for full-length cASIC1 were conducted using the Jpred4 secondary structure prediction server (*Drozdetskiy et al., 2015*).

## Cell lines

HEK293S GnTI- cells were provided by kindly provided by Dr. Gobind Khorana (*Reeves et al., 2002*) and no further authentication was conducted. Cells are routinely tested for mycoplasma contamination and all cells are mycoplasma free.

## Acknowledgements

We thank L Vaskalis for help with figures, H Owen for manuscript preparation and all Gouaux lab members for their support. Additionally, we thank Polyscope for providing the XIRAN SL 30010 polymer as a gift. This research was supported by the National Institute of Diabetes and Digestive Kidney Diseases (5T32DK007680) and the National Institute of Neurological Disorders and Stroke (5F31NS096782 to NY and 5R01NS038631 to EG). Initial electron microscopy work was performed at the Multiscale Microscopy Core at Oregon Health and Science University (OHSU). A portion of this research was performed at the National Center for Cryo-EM Access and Training and the Simons Electron Microscopy Center located at the New York Structural Biology Center, supported by the NIH Common Fund Transformative High Resolution Cryo-Electron Microscopy program (U24GM129539) and by grants from the Simons Foundation (SF349247) and New York State. Subsequent research was supported by NIH grant U24GM129547 and performed at the Pacific Northwest Cryo-EM Center at OHSU and accessed through EMSL (grid.436923.9), a DOE Office of Science User Facility sponsored by the Office of Biological and Environmental Research. Additional support was provided by ARCS Foundation and Tartar Trust fellowships. EG is an Investigator with the Howard Hughes Medical Institute.

## Additional information

### Funding

| Funder | Grant reference number | Author |
|---|---|---|
| National Institute of Neurological Disorders and Stroke | 5F31NS096782 | Nate Yoder |
| National Institute of Neurological Disorders and Stroke | 5R01NS038631 | Eric Gouaux |
| National Institutes of Health | U24GM129539 | Nate Yoder |
| National Institutes of Health | U24GM129547 | Nate Yoder |
| National Institute of Diabetes and Digestive and Kidney Diseases | 5T32DK007680 | Nate Yoder |
| Tartar Trust | | Nate Yoder |
| ARCS Foundation | | Nate Yoder |
| Howard Hughes Medical Institute | | Eric Gouaux |

The funders had no role in study design, data collection and interpretation, or the decision to submit the work for publication.

### Author contributions

Nate Yoder, Conceptualization, Data curation, Formal analysis, Funding acquisition, Validation, Investigation, Visualization, Methodology, Writing - original draft; Eric Gouaux, Conceptualization, Resources, Supervision, Funding acquisition, Methodology, Project administration, Writing - review and editing

### Author ORCIDs

Nate Yoder (iD) https://orcid.org/0000-0002-9017-0673
Eric Gouaux (iD) https://orcid.org/0000-0002-8549-2360

Decision letter and Author response
Decision letter https://doi.org/10.7554/eLife.56527.sa1
Author response https://doi.org/10.7554/eLife.56527.sa2

## Additional files

### Supplementary files
- Supplementary file 1. Cryo-EM data collection, processing and validation statistics.
- Transparent reporting form

### Data availability
The coordinates and associated cryo-EM map for the desensitized SMA-cASIC1a channel at pH 7.0 have been deposited in the Protein Data Bank and Electron Microscopy Data Bank under the accession codes 6VTK and EMD-21380, respectively. The coordinates and associated cryo-EM map for the resting SMA-cASIC1a channel at pH 8.0 have been deposited in the Protein Data Bank and Electron Microscopy Data Bank under the accession codes 6VTL and EMD-21381, respectively.

The following datasets were generated:

| Author(s) | Year | Dataset title | Dataset URL | Database and Identifier |
|---|---|---|---|---|
| Yoder N, Gouaux E | 2020 | Structure of an acid-sensing ion channel solubilized by styrene maleic acid and in a desensitized state at low pH | https://www.rcsb.org/structure/6VTK | RCSB Protein Data Bank, 6VTK |
| Yoder N, Gouaux E | 2020 | Structure of an acid-sensing ion channel solubilized by styrene maleic acid and in a desensitized state at low pH | https://www.ebi.ac.uk/pdbe/entry/emdb/EMD-21380 | Electron Microscopy Data Bank, EMD-21380 |
| Yoder N, Gouaux E | 2020 | Structure of an acid-sensing ion channel solubilized by styrene maleic acid and in a resting state at high pH | https://www.rcsb.org/structure/6VTL | RCSB Protein Data Bank, 6VTL |
| Yoder N, Gouaux E | 2020 | Structure of an acid-sensing ion channel solubilized by styrene maleic acid and in a resting state at high pH | https://www.ebi.ac.uk/pdbe/entry/emdb/EMD-21381 | Electron Microscopy Data Bank, EMD-21381 |

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
