## [Decision Letter]

**Acceptance summary:**

This study presents high-resolution structures of the resting and desensitized chicken ASIC1a channel in its native membrane environment, obtained by single-particle cryo-electron microscopy following detergent-free extraction. Whereas in previous structures of detergent solubilized ASIC1 the N-terminus of the protein was not resolved, suggesting that it is disordered, in the novel structures an N-terminal pre-membrane segment that harbors the highly conserved HG motif forms an ordered reentrant loop that lines the entire cytoplasmic half of the transmembrane pore. This arrangement provides explanations for a variety of functional observations from earlier studies, significantly advancing our understanding of ASIC structure and function.

**Decision letter after peer review:**

Thank you for submitting your article "Conserved His-Gly motif of acid-sensing ion channels, implicated in gating and ion selectivity, is a reentrant 'loop'" for consideration by *eLife*. Your article has been reviewed by three peer reviewers, including László Csanády as the Reviewing Editor and Reviewer #1, and the evaluation has been overseen by Kenton Swartz as the Senior Editor. The following individuals involved in review of your submission have agreed to reveal their identity: Youxing Jiang (Reviewer #2); Stephan A Pless (Reviewer #3).

The reviewers have discussed the reviews with one another and the Reviewing Editor has drafted this decision to help you prepare a revised submission. In recognition of the fact that revisions may take longer than we typically allow, until the research enterprise restarts in full, we will give authors as much time as they need to submit revised manuscripts.

Summary:

This study presents high-resolution structures of the resting and desensitized chicken ASIC1a channel in its native membrane environment, obtained by single-particle cryo-EM following extraction using styrene maleic acid (SMA) copolymers. The overall structures are similar to those obtained previously for detergent solubilized channels. However, whereas in the detergent solubilized structures the N-terminus of the protein was not resolved, suggesting that it is disordered, in the SMA structure an N-terminal pre-membrane segment that harbors the highly conserved HG motif forms an ordered reentrant loop that lines the entire cytoplasmic half of the transmembrane pore. This arrangement provides explanations for a variety of functional observations from earlier studies, significantly advancing our understanding of ASIC structure and function. The somewhat unexpected positioning of the conserved HG motif near the ion permeation pathway is also relevant for studies on other ENaC/DEG channel family members. In addition to scientific insight, there is also important methodological contribution. The study provides a good example of detergent-free extraction of a membrane protein directly from the cell membrane using styrene maleic acid (SMA) copolymers for structure determination.

Essential revisions:

1) Given that the SMA-assisted sample preparation has not been used on ASICs before, the reviewers have discussed whether they should request from the authors to demonstrate that the ASIC1a channel is still functional when prepared with this approach. Considering the difficulty of performing experiments in the present situation, we do not request such experiments to be done, but the authors should discuss or allude to this issue in the Discussion section.

2) The manuscript is clearly written, but the discussion is rather concise and falls somewhat short of providing enough depth and context for those not directly familiar with the ASIC literature. It would benefit from more detail with regards to the following topics:

2.1) It would be helpful if the authors could provide additional context concerning previous functional and structural data, i.e. where the present structure validates/clashes with previous observations. This is particularly evident with regards to the fact that some of the previous ASIC structures had noticeably different orientations/conformations of the TM helices. For example, in one of their previous studies (Baconguis et al., 2014), the following was stated: "A careful review of electron density maps of the initial cASIC1a structure (Jasti et al., 2007) and the delta13-PcTx1 complexes (Baconguis and Gouaux, 2012) provides no evidence for swapping of the TM2 helices as we observe in the low-pH desensitized state and in the MitTx complex. Thus, we conclude that the swap in TM2 does not occur in the low-pH and high-pH PcTx1 structures or in the initial cASIC1a structure." How do the authors rationalize the present structure in light of these earlier findings? While the construct in Jasti et al., 2007 lacked most of the N-terminus, there is no immediately obvious reason why the construct used in Bacoguis and Gouaux, 2012 would not have shown this re-entrant loop.

2.2) Related to this, could the authors speculate on why/how the residues in TM2b might impact ion selectivity? It would seem that the present structure makes it likely that the functional effects observed with mutants in this area were indirect in nature (unless of course the open state structure adopts a very different conformation in this part of the channel).

2.3) Although necessarily somewhat speculative, it would be helpful if the authors could expand on their discussion with regards to the implications of the presence of the lipids (subsection “Density features suggest TMD-membrane interact”) and the missing C-terminus (subsection “Amino terminal residues form a reentrant loop”), respectively. In other words, how do these findings compare to related (trimeric) channel structures? What could be undertaken to get more insight into these aspects? etc.

2.4) The authors suggest that the constriction formed by the conserved histidine of the HG motif might serve as a selectivity filter (subsection “Pre-TM1 residues form the lower ion permeation pathway”). Can the authors propose a chemical mechanism by which a histidine side chain would explain the ion selectivity pattern of ASIC channels?

---

## [Author Response]

Essential revisions:1) Given that the SMA-assisted sample preparation has not been used on ASICs before, the reviewers have discussed whether they should request from the authors to demonstrate that the ASIC1a channel is still functional when prepared with this approach. Considering the difficulty of performing experiments in the present situation, we do not request such experiments to be done, but the authors should discuss or allude to this issue in the Discussion section.

We have included an additional paragraph in the Discussion section to address this point.

2) The manuscript is clearly written, but the discussion is rather concise and falls somewhat short of providing enough depth and context for those not directly familiar with the ASIC literature. It would benefit from more detail with regards to the following topics:2.1) It would be helpful if the authors could provide additional context concerning previous functional and structural data, i.e. where the present structure validates/clashes with previous observations. This is particularly evident with regards to the fact that some of the previous ASIC structures had noticeably different orientations/conformations of the TM helices. For example, in one of their previous studies (Baconguis et al., 2014), the following was stated: "A careful review of electron density maps of the initial cASIC1a structure (Jasti et al., 2007) and the delta13-PcTx1 complexes (Baconguis and Gouaux, 2012) provides no evidence for swapping of the TM2 helices as we observe in the low-pH desensitized state and in the MitTx complex. Thus, we conclude that the swap in TM2 does not occur in the low-pH and high-pH PcTx1 structures or in the initial cASIC1a structure." How do the authors rationalize the present structure in light of these earlier findings? While the construct in Jasti et al., 2007 lacked most of the N-terminus, there is no immediately obvious reason why the construct used in Bacoguis and Gouaux, 2012 would not have shown this re-entrant loop.

Given the history of multiple conformations – i.e. domain swapped vs continuous – at the TM2 helix of cASIC1 x-ray structures as well as the subject matter of this work, we recognize that a more comprehensive overview of this disparity is warranted. To this end, we have provided a synopsis of TM heterogeneity in cASIC1 structural studies in the Discussion section as well as our interpretation of these past results in light of the present study. Briefly:

With regards to the conformation of TM2 helices (Discussion section): While domain-swapped TM2 helices and GAS belt constrictions are now established structural characteristics of functional channels in desensitized, toxin-stabilized open, and resting conformations, x-ray structures of ASIC1 channels bound to the gating modifier toxin, PcTx1, harbor continuous TM helices, indicating that a subset of conformational states outside of the canonical proton-dependent gating pathway may deviate from the domain-swapped architecture. While the capacity of ASICs to adopt both domain-swapped and continuous TM2 helices is supported by both x-ray structures and evolutionary analysis, the functional relevance of a ‘dual-mode’ lower pore architecture remains enigmatic.

With regards to the pre-TM1 reentrant loop (Discussion section): In light of the structural heterogeneity at the lower pore of ASICs, the presence of lipid-like densities along the reentrant loop of cASIC1-SMA channels is particularly notable. Indeed, these studies represent the first structures of an ASIC not solubilized by detergent. We speculate that a lipid-like environment, maintained by virtue of detergent-free, SMA-mediated protein extraction and purification methods, may be important for maintaining the integrity of the lower pore structure of ASIC1 channels. Accordingly, structures of the cASIC1-SMA complex provide evidence of an ordered pre-TM1 reentrant loop in the high pH resting and low pH desensitized conformations.

2.2) Related to this, could the authors speculate on why/how the residues in TM2b might impact ion selectivity? It would seem that the present structure makes it likely that the functional effects observed with mutants in this area were indirect in nature (unless of course the open state structure adopts a very different conformation in this part of the channel).

We have expanded the Discussion section to include an overview of the potential contribution of TM2b residues to selectivity in ASICs. As outlined in the comment, we acknowledge that the structure of a full-length cASIC1 channel in a lipid-like environment and in a proton-activated open state may adopt a different conformation than indicated by existing structures. Currently, however, our structural data does not provide evidence of a direct contribution made by TM2b residues to the ion channel pore and thus we conclude that the effects of TM2b mutations are likely indirect in nature, such as may be observed if mutations at TM2b altered the conformation of the lower pore.

2.3) Although necessarily somewhat speculative, it would be helpful if the authors could expand on their discussion with regards to the implications of the presence of the lipids (subsection “Density features suggest TMD-membrane interact”) and the missing C-terminus (subsection “Amino terminal residues form a reentrant loop”), respectively. In other words, how do these findings compare to related (trimeric) channel structures? What could be undertaken to get more insight into these aspects? etc.

We believe that the presence of elongated lipid-like densities, especially adjacent to the reentrant loop, is indicative of how our detergent-free protein handling methods retained a lipid-like environment surrounding the TMD and protected the integrity of the reentrant loop residues. We believe that this hypothesis supports our observation of disordered pre-TM1 reentrant loop residues in x-ray and cryo-EM structures of detergent-solubilized channels and have expanded the Discussion section to include these points. We also discuss the disordered carboxy terminus and the contrast to recent structures of trimeric P2X_7_ channels, which harbor large structured intracellular terminal domains (Discussion section), concluding that ‘Contrary to P2X receptors, the intracellular carboxy termini of ASICs is predicted to be largely disordered, does not appear to contribute to channel gating and is instead likely to serve as a nexus for interactions with cytoplasmic proteins, thus highlighting the need for future studies to pursue the structure of ASICs in complex with cellular binding partners in an effort to better illuminate the contribution of these residues to ASIC biology.’

2.4) The authors suggest that the constriction formed by the conserved histidine of the HG motif might serve as a selectivity filter (subsection “Pre-TM1 residues form the lower ion permeation pathway”). Can the authors propose a chemical mechanism by which a histidine side chain would explain the ion selectivity pattern of ASIC channels?

We have expanded on our initial hypothesis of His 29 may contribute to ion selectivity of ASICs and have included this in the Discussion section. However, while the presence of a previously unobserved constriction in the pore of cASIC1 in intriguing and may indicate involvement in selectivity, these concepts are, at this stage, speculative given the limitations of our current data. We hope that future studies will address the lack of a structure of a proton-activated (open) ASIC in a lipid-like environment as it will provide insight to the conformation adopted by the lower pore and reentrant loop when in an activated and ion conducting conformation.